# Regional Conservation Assessment of the Threatened Species: A Case Study of Twelve Plant Species in the Farasan Archipelago

Rahmah N. Al-Qthanin [1,2,*] and Samah A. Alharbi [3]

1   Biology Department, College of Sciences, King Khalid University, Abha 62529, Saudi Arabia
2   Prince Sultan Bin-Abdul-Aziz Center for Environmental and Tourism Studies and Research, King Khalid University, Abha 61421, Saudi Arabia
3   Department of Biology, Faculty of Applied Science, Umm Al-Qura University, Makkah 24381, Saudi Arabia
*   Correspondence: rngerse@kku.edu.sa

**Abstract:** Assessing species at the regional level for their conservation is a vital first step in identifying and prioritizing species for both ex situ and in situ conservation actions. The complex coastal geomorphology of the Farasan Archipelago gives rise to promontories and bays that fragment the coastal flora. Climate change studies, combined with a case study of anthropogenic land use changes such as urbanization, tourism, and fishing, highlight the threat to the fragmented plant populations. In this study, the regional IUCN categories and criteria have been used to assess the conservation status of twelve targeted taxa of the Farasan Archipelago based on the data collected during field surveys and a literature review. According to our results, six species have been categorized as endangered (EN), four species as vulnerable (VU), and two species as near threatened (NT). Compared to an earlier assessment at the global level, *Avicennia marina* and *Rhizophora mucronata* have been re-categorized with a high degree of threat and ten species have been assessed for the first time. An effective action plan for the protection of the coastal zone and inland area biodiversity of the Archipelago is crucial to reduce threats to the islands' plants.

**Keywords:** Farasan Archipelago; Regional Red List; conservation; threatened species; coast zone

## 1. Introduction

The Farasan Archipelago is an Important Plant Area in the Arabian Peninsula [1], and is the first Saudi reserve ever registered with UNESCO Scientific Man and the Biosphere Program in 2021 [2]. It is the second largest Red Sea Archipelago after the Archipelago of Dahlek (Eritrea) [3] and has high priority in terms of conservation [4].

The interior surface of the islands is a subtropical desert of fossil limestone [5], interspersed with many short water runnels that provide fertile farmland. The vegetation cover is sparse except in ravines between fossil coral outcrops, which have the highest species richness and the greatest number of annuals [5,6]. These outcrops are a concentrated pool of individual species that are rare elsewhere in the Arabian Peninsula and the Red Sea area. Most of them are located in the main Islands (Farasan Alkabir, 381 km$^2$ and Sajid, 149 km$^2$).

Over the past two decades, the Farasan Archipelago have been recorded as the only Saudi locality for a total of 14 species, including *Basilicum polystachyon*, *Dinebra retroflexa*, *Dinebra somalensis*, *Euphorbia collenetteae*, *Ficus populifolia*, *Flueggea leucopyrus*, *Indigofera semitrijuga*, *Ipomoea hochstetteri*, *Limonium cylindrifolium*, *Micrococca mercurialis*, *Nothosaerva brachiata*, *Rorida brachystyla*, *Taverniera cuneifolia*, *Vahlia digyna* [1,7–9]. However, the revision of available floristic publications of Saudi Arabia [10–20] showed that eight of which have been found in other localities in the mainland of Saudi Arabia (Appendix A, grey shaded species). More recently, two species added to the flora of Saudi Arabia known so far from the Farasan Islands: *Blepharis saudensis* [21] and *Indigofera cordifolia* [22]. The significance

of the Farasan Islands flora in Saudi Arabia, thus, is not in terms of its endemism, which is low, but in the presence of those eight species rare elsewhere in the Arabian Peninsula (hereafter, Farasan restricted species).

The coastal zone is comprised of a mix of natural beaches that are rocky, sandy (fine or coarse coral fragments), or vegetated [9,23] as seen in Figure 1. The coastal area of the Farasan Archipelago is a major source of income for local inhabitants and communities [24]. It is an area of national and international significance for breeding seabirds, shorebirds, and marine mammals in the Red Sea, including the endemic *Larus leucophthalmus* and *Dromas ardeola* [25]. It is the site for the unique annual aggregation of parrotfish [26]. To coincide with this aggregation, the Al Harid Festival occurs in March or April every year in one of the inner bays of Farasan Alkabir Island. The parrotfish, in large numbers, move into the shallow water of the bay where people can easily harvest them in large quantities by wading into the water with nets [26]. In addition, it supports a large number of marine, mangrove, and wetland species [27].

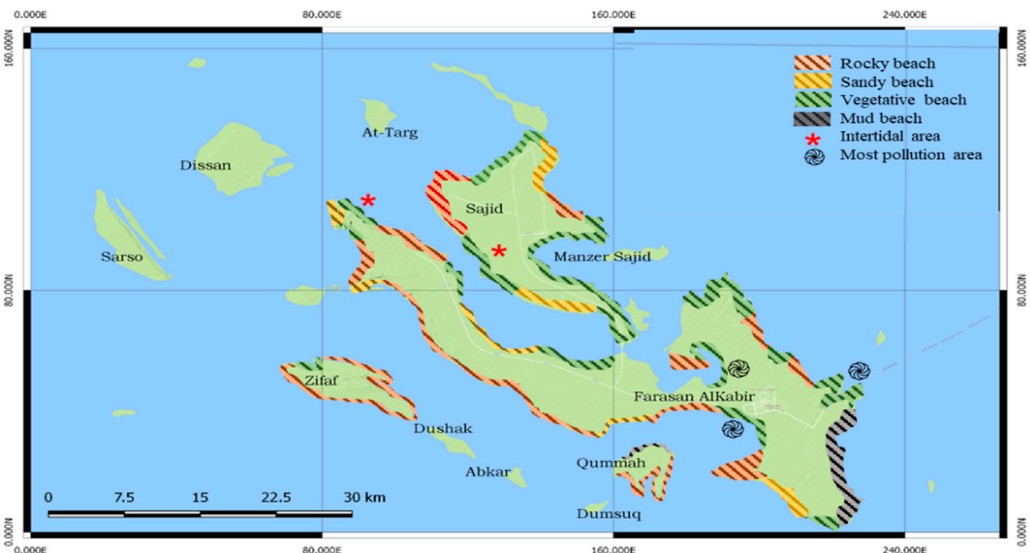

**Figure 1.** The vegetative beach in the Farasan Archipelago. A spiral shows the most pollution area along the coast zone and asterisks shows the intertidal zone in the Farasan Archipelago.

Threats to the island ecosystem have developed over recent decades, and Figure 2 shows some of those threats. The main Islands (Farasan Alkabir and Sajid) are undergoing rapid change in economic infrastructure [28], and population growth [29] that overcome the natural habitats in these islands. The excessive grazing and agricultural expansion have increased the threat to the local biodiversity of the island's interior [9]. Moreover, the invasive *Prosopis juliflora* has spread to and negatively affected the growth of the native *Vachellia flava* woodlands [1,30]. These woodlands are the main dietary source of the vulnerable endemic subspecies *Gazella gazella farasani* [31] and therefore play an important role in the conservation of this gazelle.

The rise in sea level [32] due to the melting of frozen water mainly in the Antarctic and Arctic regions [33] has caused changes to the coastlines and extent of the islands. The sea level rise in the Farasan Archipelago is predicted to be between 0.18 m to 1.2 m by 2100 [34]. As the sea level rises, coastal habitats are inundated, eroded, or washed away, which can result in habitat loss [35].

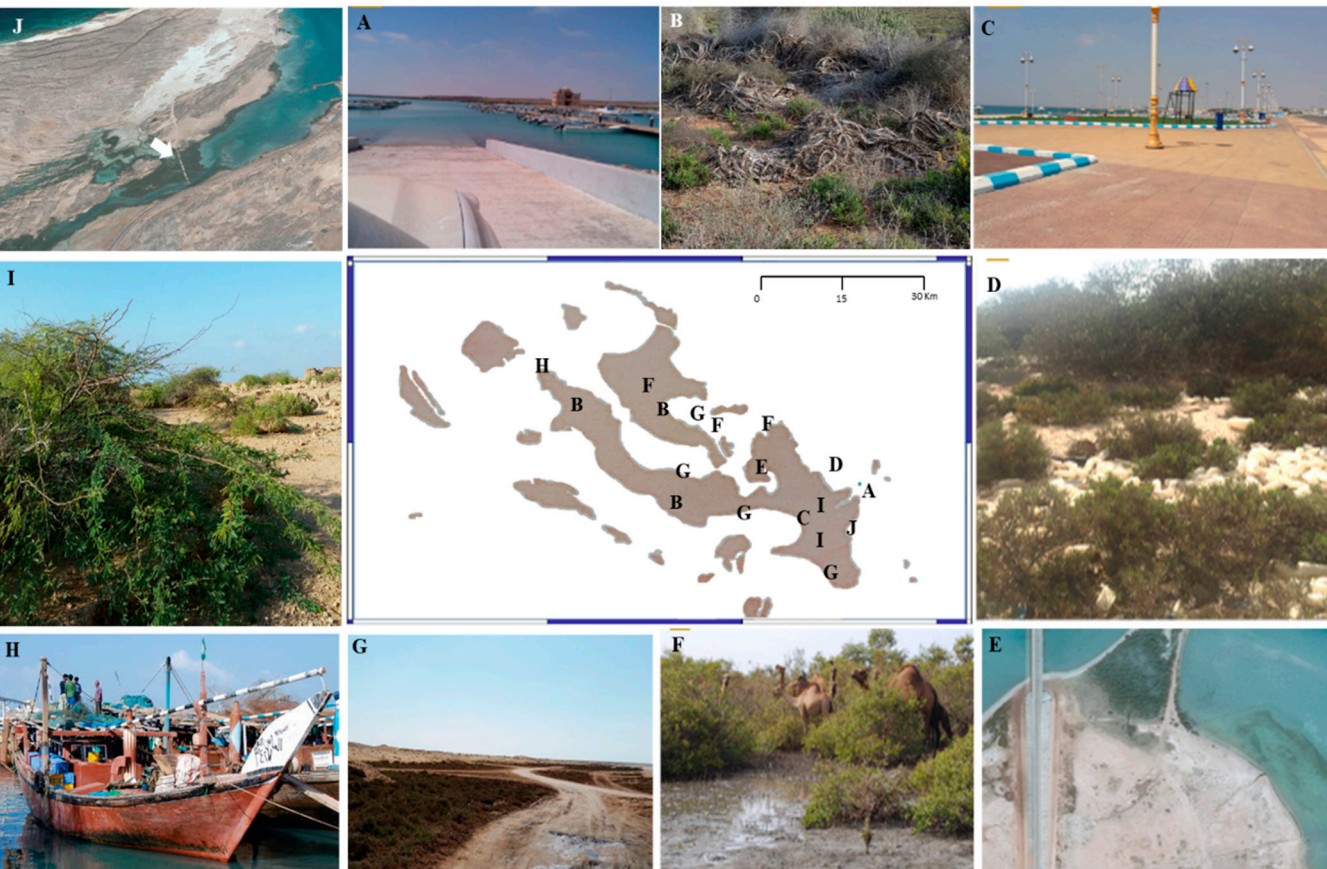

**Figure 2.** The Farasan Archipelago map of the most threatened areas. (**A**) The Farasan Island port, (**B**) removed *Euphorbia collenetteae* to convert land for agricultural use and housing construction, (**C**) Jannabah beach gardens, (**D**) waste disposal in *A. marina* area, (**E**) off-road around and within *A. marina* population behind Sajid Island Bridge, (**F**) camel overgrazing (SWA), (**G**) off-road around along low-shoreline (mostly of *Tetraena* spp. habitat). (**H**) Small boats in a fishing area, (**I**) the invasive *Prosopis juliflora* penetrating *Vachellia flava* woodland in Al-Muharraq area, south-east Farasan Alkabir. (**J**) New road by the sand dividing the mangroves area into two areas, closing the water channels to the internal part. The authors took all photos except E & J.

The coastline has suffered vast changes in land use through anthropogenic factors, including urbanization and sand extraction, fishing, shipping, pollution, military purposes, tourism, and off-road driving. The development of beach resorts, particularly on Farasan Alkabir Island [27] and the sandy beach areas, admit special problems due to the removal of beach sand for local construction projects. This has destroyed the low-lying shoreline vegetation [4] and has disrupted shoreline dynamics and obstructed tidal flows [36]. Between 59 and 76% of artisanal fishing occurs on the bays of the Farasan Archipelago [27]. In addition, cutting down the mangroves for the purpose of making traps for catching migratory birds on the uninhabited islands [9] has led to a loss of the mangrove ecosystem.

The Farasan Archipelago is a major international shipping route, used for an estimated 25,000–30,000 ship transits annually [26]. This means that the coastal biodiversity is subject to oil pollution. Most sewage is dumped inland, with disposal into the sea occurring only at one site inside Farasan Port.

Expansion has happened in the tourism sector, particularly with the new vision of the government of Saudi Arabia by 2030 [37]. Around 4000 to 5000 domestic tourists visit the Farasan and Sajid Islands during school vacations, putting added pressures on the sewage system, water demands, and through activities such as recreational off-road driving, which damages vegetation and disrupts the soil surface.

All of these threats affect the connectivity between plant populations [26], making the maintenance of viable species more difficult. Mangroves, *Avicennia marina* and *Rhizophora mucronata* [38], *Tetranea* spp. (previously known as *Zygophyllum* spp.) [27], and the nationally rare species [9] are particularly prone to being threatened by these factors. Moreover, mangrove species are, both regionally and globally threatened [1,36]. These species may help drive the conservation of the coastal island ecosystem.

In recent years, the IUCN Red List categories and criteria have been increasingly used at the regional level. A first attempt to make the IUCN regional guidelines work at a regional level was made by the Regional Application Working Group (RAWG) [39], after which they received many suggestions to amend the guidelines and to test them in real situations [40]. A final version of the regional guidelines was published by IUCN in 2012 [41].

Although the Farasan Archipelago was established as a protected area in 1989 [42] and an Important Plant Area in 2010 [1], the conservation status of the more vulnerable species has had only a limited amount of attention compared to the Socotra Archipelago and Carrabin Islands [38,43]. There has been no previous regional IUCN Red List published available on the Farasan Archipelago.

The Regional Red List assessments are important in order to monitor the status of the biodiversity of the taxa at a regional level, and this may theoretically prevent or delay species extinction globally [44]. The increase in the public awareness of the human impact on biodiversity affects the realities of conservation planning and funding [45]. The aims of this research are: (1) to assess conservation status and to produce a red list of twelve exemplar species in the Farasan Archipelago; (2) to provide an analysis and information on the status of those species, and on any trends and threats in order to inform and catalyze actions for biodiversity conservation; (3) to create a reference and baseline for a series of studies for the assessment of the other species in the critical habitat of the Farasan Islands.

## 2. Material and Methods

Twelve species were chosen to represent the threatened species and habitats of the Farasan Archipelago for the regional conservation assessment (Table 1). Six of which represent the coastal areas vegetation: two mangrove plants and four species of Zygophyllaceae. The remaining six are the nationally rare restricted species to the Farasan Islands: one Lamiaceae, one Poaceae, two Euphorbiaceae, one Cleomaceae, and one Vahliaceae. *Blepharis saudensis* and *Indigofera cordifolia* were excluded as they have recently been assessed as Endangered and Nationally Endangered, on the basis of the IUCN criteria [21,22], respectively. Names were updated following the World Flora Online and relevant taxonomic publications, details can be found in Table 1.

Three field trips were conducted to the Farasan Archipelago in April 2016 and December 2016/2017, covering six islands: Farasan Alkabir, Sajid, Qummah, Zifaf, Dumsuk, and Dawshak. Field observations were recorded, including, habitat conditions, species distribution data, threats were evident, and interviews with local people. Further data including (habitat, ecology, population, uses, and any threats of species) were gathered through previously published studies. Additional data were also collected from herbarium records. The locality and habitat information was derived from specimen labels seen at the Royal Botanic Garden, Kew (K) http://apps.kew.org/herbcat/navigator.do, accessed on 1 June 2019 and the Royal Botanic Garden, Edinburgh (RBGE) https://data.rbge.org.uk/search/herbarium/, accessed on 1 June 2019.

The combined data from all three sources and any updates in distribution based on recent data were used to provide updated distribution maps of the species created by Arcview GIS software [46] and GeoCAT software [47]. The extent of occurrence (EOO) was calculated by constructing the minimum convex polygon around known occurrences, and the area of occupancy (AOO) was calculated by overlaying a grid of $2 \times 2$ km$^2$ and counting the occupied grid cells [48]. If the EOO was less than the AOO, the EOO was changed to make it equal to the AOO to ensure consistency with the definition of the AOO as an area within the EOO following the IUCN guideline recommendation [49].

After data were gathered, this study followed IUCN criteria and categories version 15.1 [48] and the regional IUCN guidelines version 4.0 [41], to do the Red listing. The classification obtained from the IUCN assessment was adjusted based on the status of the species populations outside of the country. Depending on whether the outside populations could pose a rescue effect on the risk of extinction of the national population, a downgrade, upgrade, or no change was applied to the categories [41].

**Table 1.** Summary of the nomenclature changes of the species under study (updates, references, family).

| Species Name | Currently Accepted Name | References | Family |
|---|---|---|---|
| *Avicennia marina* (Forssk.) Vierh. | *Avicennia marina* (Forssk.) Vierh. | [50] | Acanthaceae |
| *Rhizophora mucronata* Poir. | *Rhizophora mucronata* Poir. | [51] | Rhizophoraceae |
| *Zygophyllum simplex* L. | *Tetraena simplex* (L) Beier and Thulin | [52] | Zygophyllaceae |
| *Zygophyllum album* L.f. | *Tetraena alba* var. *alba* (L.f.) Beier and Thulin | [52] | Zygophyllaceae |
| *Zygophyllum coccineum* L. | *Tetraena coccinea* (L) Beier and Thulin | [52] | Zygophyllaceae |
| *Zygophyllum propinquum* ssp. *migahidii* (Hadidi) Thomas and Chaudhary | *Tetraena propinqua* (Decne.) Ghazanfar and Osborne ssp. *migahidii* (Hadidi ex Beier and Thulin) Alzahrani | [53,54] | Zygophyllaceae |
| *Basilicum polystachyon* (L.) Moench | *Basilicum polystachyon* (L.) Moench | [7,55] | Lamiaceae |
| *Drake-Brockmania somalensis* Stapf | *Dinebra somalensis* (Stapf) P.M.Peterson and N.Snow | [56] | Poaceae |
| *Euphorbia collenetteae* D.Al-Zahrani and El-Karemy | *Euphorbia collenetteae* D.Al-Zahrani and El-Karemy | [57] | Euphorbiaceae. |
| *Micrococca mercurialis* (L.) Benth. | *Micrococca mercurialis* (L.) Benth. | [58] | Euphorbiaceae |
| *Cleome noeana* Boiss. subsp. *brachystyla* (Deflers ex Franch.) D.F.Chamb. and Lamond | *Rorida brachystyla* (Deflers ex Franch.) Thulin and Roalson | [59] | Cleomaceae |
| *Vahlia digyna* (Retz.) Kuntze | *Vahlia digyna* (Retz.) Kuntze | [60] | Vahliaceae |

## 3. Results and Discussion

### 3.1. Avicennia marina (Forssk.) Vierh.

**Family:** Acanthaceae

**Synonyms:** *Avicennia alba* Blume; *Avicennia alba* var. *latifolia* Moldenke; *Avicennia intermedia* Griff.; *Avicennia marina* var. *alba* (Blume) Bakh.; *Avicennia marina* f. *angustata* Moldenke; *Avicennia marina* var. *anomala* Moldenke; *Avicennia marina* var. *intermedia* (Griff.) Bakh.; *Avicennia marina* f. *intermedia* (Griff.) Moldenke; *Avicennia marina* subsp. *marina*; *Avicennia marina* var. *marina*; *Avicennia mindanaensis* Elmer; *Avicennia officinalis* var. *alba* (Blume) C.B.Clarke; *Avicennia sphaerocarpa* Stapf ex Ridl.; *Avicennia spicata* Kuntze; *Avicennia tomentosa* var. *arabica* Walp.; *Sceura marina* Forssk.; *Avicennia balanophora* Stapf and Moldenke ex Moldenke.

**Common name**: Grey Mangrove, White Mangrove, and Tivar.

**Local Names in the Farasan Islands:** Qurm, Gurm and Shorah.

**Geographic distribution range:** It spreads in the intertidal mudflats which have extremely limited wave action below the high watermark along the shores of the seas and oceans [61]. It is distributed through the western Indian Ocean, including Madagascar and Mozambique, northwards to Egypt and Saudi Arabia where it occurs on the coast of the Red Sea [62].

**Distribution (countries of occurrence):**

South-west, south, and south-east Asia, Australia, and northern parts of New Zealand. It is one of the few mangroves found in the arid regions of the coastal Arabian Peninsula, mainly in sabkha environments in the United Arab Emirates, Qatar, Bahrain, Oman, as well as in similar environments on both side of the Red Sea (in Yemen, Saudi Arabia, Egypt, Eritrea, and Sudan) and Qatar and southern Iran along the Persian Gulf coast. It

is a characteristic species of the Southern Africa mangroves ecoregion and is one of three species present in Africa's southernmost mangroves.

**Biology:**

It is a shrub up to a medium-sized tree; 2–5 m tall. It has an extensive underground root system with pneumatophores up to 9 cm long, sticking up out of the mud in dense stands spreading out from tree. The leaves are in opposite pairs, thick, leathery, shiny olive green above, with a margin that is entirely sharp or with a bluntly pointed tip, with the base narrowing and a short petiole around 5 mm long [12,63].

**Flowering and Fruiting period:**

The flowers are creamy yellow, small, in dense round heads in leaf axils or terminally. The fruit is green and oval. The flowering and fruiting time is February–June; March–August [63].

**Reproduction:**

Reproduces by seeds, these developing on the tree, with the fruit usually splitting after falling. The seed is water-dispersed [12].

**Habitat and Ecology:**

shore-line habitat

**Population information**

The population trend in Zifaf and Sajid Islands is probably stable because it has the ability to re-colonize disturbed sites when the environment becomes favorable again, due to its effective dispersal mechanism. However, in Farasan Alkabir Island, there has been a significant local degradation of coastal habitats associated with the growth of domestic tourism [24,27,63–67]. Currently, this Island has the largest area of suitable habitat and largest populations of the species. These threats are due to habitat loss caused by urban and industrial development along the coast, specifically near the port of Farasan Alkabir where a large population has been subjected to massive human activities, such as the construction of the sea port, highway, mersa, and a side road across the khor, leading to destruction on a massive scale [68].

**Threats:**

Infrastructure development related to transport, such as roads and bridges, has also caused damage. Many of the populations of *A. marina* on Sajid Island have been lost due to construction of a bridge connecting Farasan Alkabir and Sajid Island [69]. There is evidence of mortality in a large number of *A. marina* on the other side of the port where engineering work has prevented water flow [68]. Many sand dams were created, which closed the water channels for several *A. marina* populations. Local people harvest this species for medicinal uses, such as treating skin diseases in folk medicine. This suggests that it possesses some natural antimicrobial, anti-bacteriophage, and cytotoxic activities [70]. Wood is often used for fuel [47].

In two localities, Farasan Alkabir (Al Qandal area) and Zifaf Island, this species shares the same shore-line habitat with *Rhizophora mucronata* and they are seen growing side by side. Pollution from sewage has been recorded, especially in relation to the port area and over-fishing activities. Browsing and trampling by camels, gazelle, and goats causes habitat degradation [26]. These threats are ongoing and increasing, especially given the new plans for recreation on the islands [71].

**Criteria applied:**

The category of *A.marina* is Endangered (EN), then it has been downlisted to Vulnerable (VU) at the regional level due to the presence of the populations of this species in neighboring countries and islands. The extent of occurrence (EOO) is 419 km$^2$. The area of occupancy (AOO) is estimated to be 44 km$^2$ (possibly ranging from 40–60 km$^2$) (Figure 3). However, the probability of immigration from neighboring locations is unknown.

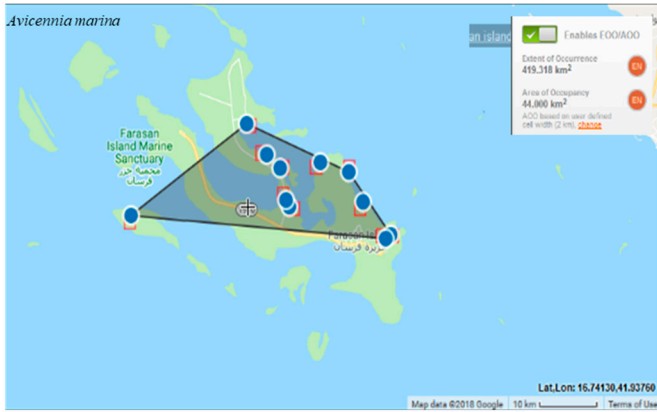 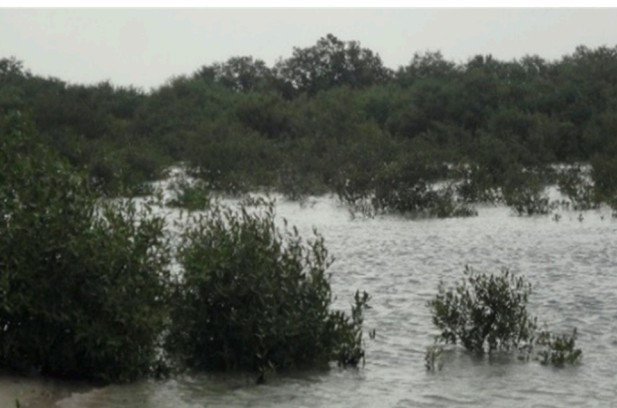

**Figure 3.** Geographical distribution of *Avicenna marina* in the Farasan Archipelago (plant photo by R. Al-Qthanin from Faranan Alkabir Island).

B1ab (i, ii, iii, v) +2ab (i, ii, iii, v)
**Previous assessments (and global assessments):**
Least Concern (LC)
**Conservation actions needed:**
It should hold special importance in conservation terms of this species. Conservation measure of this species should include planting, management for sustainable use, and legal protection of some areas of islands.

### 3.2. Rhizophora mucronata Poir.

**Family:** Rhizophoraceae
**Synonyms:** *Mangium candelarium* Rumphius; *Rhizophora candelaria* Wight and Arn; *Rhizophora longissima* Blanco; *Rhizophora macrorrhiza* Griff.; *Rhizophora longissima* Blanco; *Rhizophora mangle* Roxb. (non-L.); *Rhizophora mucronata* f. *reducta* Hochr.; *Rhizophora rugens* Ehrenb. ex. Schweinf.
**Common name:** Mangrove, Red Mangrove, Seebasboom, and Asiatic Mangrove.
**Local Names in the Farasan Islands:** Kendal.
**Geographic distribution range:**
Globally, it occurs along the intertidal regions of tropical and sub-tropical coasts [72].
**Distribution (countries of occurrence):**
Australia; Bangladesh; Brunei Darussalam; Cambodia; Comoros; Djibouti; Egypt; Eritrea; India; Indonesia; Iran, Islamic Republic of; Japan; Kenya; Madagascar; Malaysia; Maldives; Mauritius; Mayotte; Micronesia, Federated States of Mozambique; Myanmar; Oman; Pakistan; Palau; Papua New Guinea; Philippines; Réunion; Saudi Arabia; Seychelles; Singapore; Solomon Islands; Somalia; South Africa; Sri Lanka; Sudan; Taiwan, Province of China; Tanzania, United Republic of Thailand; United Arab Emirates; Vanuatu; Viet Nam; Yemen.
**Biology:**
A small to medium-sized tree starting from 2–5 m and growing up to 10 m tall, with strong apical dominance and distinctive aerial roots which are rough and reddish. The leaves are compact, simple, opposite, broadly elliptic to oblong-elliptic, leathery, hairless, glossy, dark green to yellowish green, crowded towards the end of branches, and with smooth margins with a pointed apex [13].
**Flowering and Fruiting period:**
It has creamy white flowers, with a few arranged in the form of axillary heads. The fruit is single seeded and up to 70 mm long, germinating while still on the tree (viviparous) [13]. The flowering and fruiting time as following February–June; March–August.
**Reproduction:**
Reproduces by seeds, these developing on the tree, with the fruit usually splitting after falling. The seed is water-dispersed [13].

**Habitat and Ecology:**

Shore-line habitat.

**Population information:**

Habitat loss is due to erosion in the available habitats, urbanization, and a side road put down by Saudi Wildlife Authority SWA and the Border Guards, as it is located at the edge of Farasan Al Kabir. This species has a more limited distribution than *A. marina*, this may be because it is at the edge of its natural climatic distribution [63].

**Threats:**

It is threatened by overgrazing and tourism. The population trend is near- stable, with a limited distribution [63]. Local people harvest this species for medicinal uses, such as to treat angina, diabetes, diarrhoea, dysentery, hematuria, and haemorrhage [73]. The wood is also used for fuel and for building ships due to the high quality [74].

**Criteria applied):**

The regional extent of occurrence (EOO) and the area of occupancy (AOO) are esti-mated to be 16 km$^2$ (Figure 4). The category has been downlisted from Endangered (EN) to Vulnerable (VU) at the regional level, because of the presence of this species on neighboring Islands. EN is then downlisted to VU because of the presence of populations outside the islands.

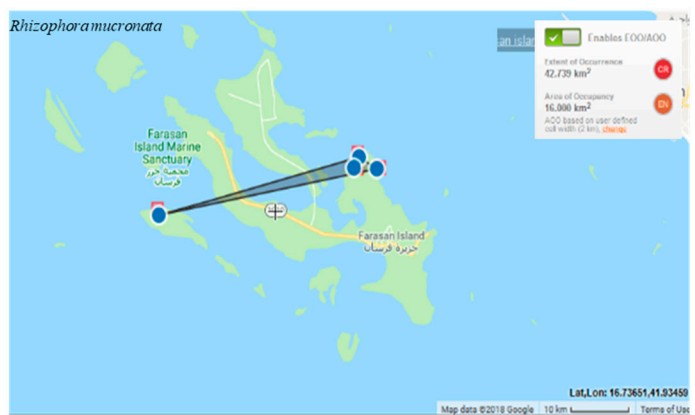 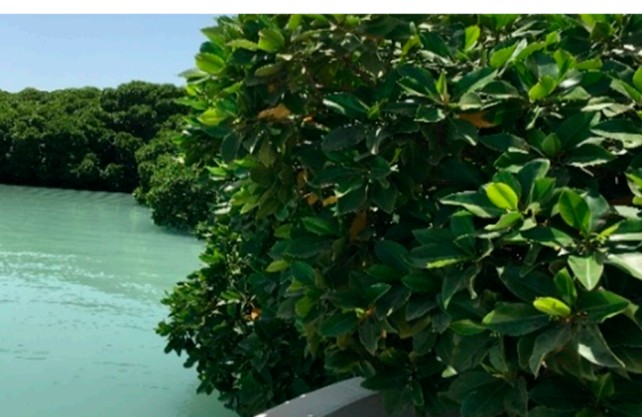

**Figure 4.** Geographical distribution of *Rhizophora mucronate* in the Farasan Archipelago (plant photo by R. Al-Qthanin from Faranan Alkabir Island).

B1ab (i, ii, iii, v) +2ab (i, ii, iii, v).

**Previous assessments (and global assessments):**

LC

**Conservation actions needed:**

It should hold special importance in conservation terms of this species. Conservation measures of this species should include planting, management for sustainable use, and legal protection of some areas of islands.

*3.3. Tetraena simplex (L.) Beier and Thulin*

**Family:** Zygophyllaceae

**Synonyms:** *Zygophyllum Simplex* L., Mant. Pl. 68 (1767); *Zygophyllum portulacoides* Forssk. Fl. Egypt. arab.:88 (1775); *Fabago portulacifolius* Medik. *Zygophyllum dregeanum* C.Presl; *Zygophyllum microphyllum* Eckl. and Zeyh. *Zygophyllum obtusum* Vicary; *Zygophyllum portulacoides* Forks. *Zygophyllum simplex* var. *herniarioides* Chiov.

**Common name:** Brakkies, Brakspekbos, Brakspekbossie, Panspekbos, Rankspekbos, Volstruisdruiwe, Volstruis-slaai.

**Local Names in the Farasan Islands:** Harm, Om thoreyb, Hamd, and Qarmal.

**Geographic distribution range:**

This species is not endemic to the Farasan Islands. It has a provincial distribution from the Mediterranean through to Central Asia, South Africa, and Australia [13].

**Distribution (countries of occurrence):**

West Asia and Africa. It can be found as far east as India. The most common habitats are shrub-steppes and deserts, and it grows best in salty conditions.

**Biology:**

*Tetraena simplex* differs from other *Tetraena* species in some of its morphological characteristics. The species is an annual herb, and the leaves are simple and sessile. The color of the flower is yellow, the staminal appendages are bipartite, and the fruit shape is obovoid and 5-lobed [13,75]. It has been used traditionally to treat gout, asthma, and inflammation [76–78].

**Flowering and Fruiting period:**

The flowering and fruiting time as following August–May; October–November.

**Reproduction:**

Reproduces by seeds.

**Habitat and Ecology:**

Shore-line habitat-soil salinity.

**Population information:**

High soil salinity is probably the cause of the low species density in this area. Rain, inundation by the sea, and the depth of the water table plays a prominent role in regulating the community of this species [79]. The habitat is sandy and has degraded because of the sand removal for urbanization and the development of gardens [27].

**Threats:**

The dramatic loss of habitat leads to the fragmentation and isolation of *T. simplex*. Livestock overgrazing, escalating sand mining activities, and the demand for sand by new development schemes can lead to the disappearance of some of the smaller beaches [69].

**Criteria applied:**

The regional extent of occurrence (EOO) is 826 km$^2$. The area of occupancy (AOO) is estimated to be 555 km$^2$ (Figure 5). This results in the categorization of Near Threatened (NT), because of the presence of this species on neighboring islands. VU is then downlisted to NT because of the presence of populations outside the islands.

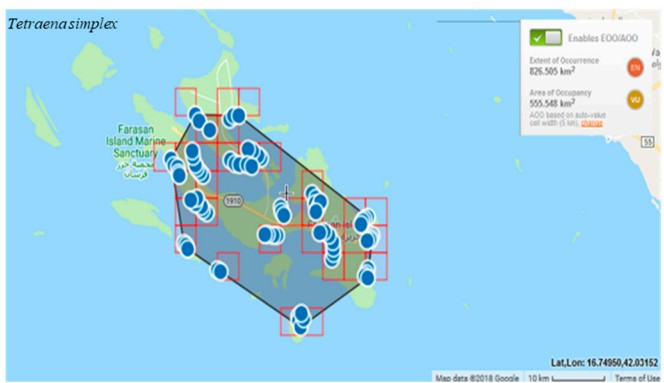 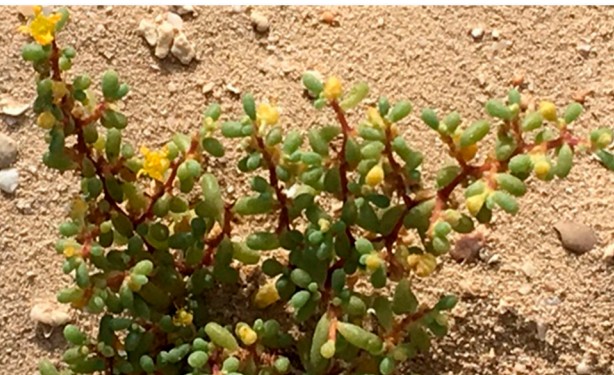

**Figure 5.** Geographical distribution of *Tetraena simplex* in the Farasan Archipelago (plant photo by R. Al-Qthanin from Faranan Alkabir Island) .

B1ab (ii, iii, v) +2ab (ii, iii, v).

**Previous assessments (and global assessments):**

Has not yet been assessed.

**Conservation actions needed:**

The off-road traffic is the leading contributor to habitat degradation in the main islands, Farasan Alkabir and Sajid, and one of the major threats to species. Management policies need to be directed towards controlling off-road vehicle driving, restoring the affected areas, and maintaining ecosystem function in these islands.

### 3.4. Tetraena alba var. alba (L.f.) Beier and Thulin

**Family:** Zygophyllaceae

**Synonyms:** *Zygophyllum album* L.f.; *Zygophyllum album* var. *amblyocarpum* (Bak. fil. ex Oliv.) Hadidi; *Zygophyllum amblyocarpum* Bak. Fil. *Zygophyllum proliferum* Forsk.

**Common name:** Weißes Jochblatt (DE); White Bean-caper (EN)

**Local Names in the Farasan Islands:** Rotreyt, Qarmal, Harm.

**Geographic distribution range:**

Its native range extended from northeast Spain to Arabian Peninsula and Somalia [54].

**Distribution (countries of occurrence):**

Egypt, Jordan, Tunisia, Palestine, Somalia, South Africa, and Greece, and Saudi Arabia (Farasan Islands) [13,75].

**Biology:**

The species is small shrub, perennial, and the stem is green or greenish grey. The leaves are fleshy, 2-foliolate cylindrical with acute apex. The color of the flower is white, arranged in clusters. The fruit shape is obconical, 5-ridged at the upper end [75].

**Flowering and Fruiting period:**

The flowering and fruiting time as following August-May; October–November.

**Reproduction:**

Seed.

**Habitat and Ecology:**

The habitat of this species is coastal and inland on saline sandy soils, sand dunes and plains, and saline depressions [80].

**Population information:**

This habitat is subject to loss and consequent fragmentation, resulting in the isolation of the remaining communities. These fragmented vegetation patches grow in high soil salinity on exposed shorelines, suffering heat and strong winds in open coastal areas [81].

**Threats:**

Human factors have a main impact on the coastal habitat that leads to the area being unable to provide conditions that can ensure the continued viability of the species. This species has a very limited distribution. Medicinally, it is used for hypertension complications [82]. It is used in traditional medicine as a remedy for rheumatism, gout, hypoglycaemia, and as anti-eczema treatment [83].

**Criteria applied:**

The regional extent of occurrence (EOO) is 7 km². The area of occupancy (AOO) is estimated to be 7 km² (Figure 6). It is Critically Endangered locally (CR) but possible migration from neighboring countries results in a downlisted threat category to endangered EN.

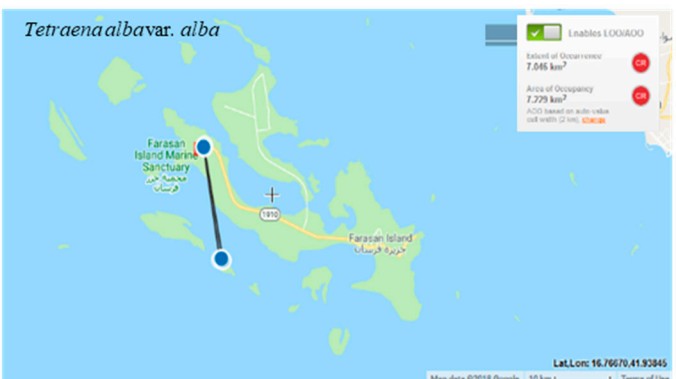 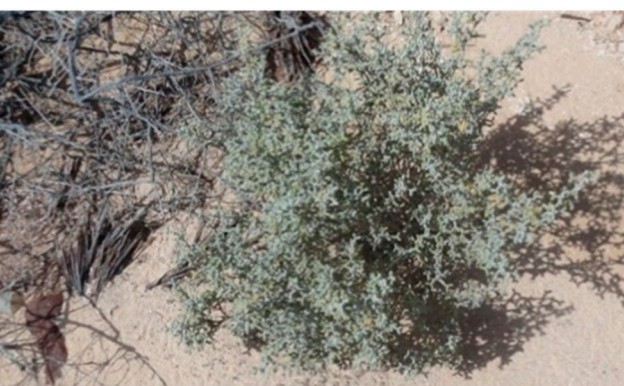

**Figure 6.** Geographical distribution of *Tetraena alba* var. *alba* in the Farasan Archipelago (plant photo by R. Al-Qthanin from Dushak Island).

B1ab (ii,iii,v) + 2ab(ii,iii,v); C2a(i);D.

**Previous assessments (and global assessments):**

Has not yet been assessed.

**Conservation actions needed:**

The off-road traffic is the leading contributor to habitat degradation in the main islands, Farasan Alkabir and Sajid, and one of the major threats to species. Management policies need to be directed towards controlling off-road vehicle driving, restoring the affected areas and maintaining ecosystem function in these islands.

*3.5. Tetraena coccinea (L.) Beier and Thulin*

**Family:** Zygophyllaceae

**Synonyms:** *Zygophyllum berenicense* (Muschl.) Hadidi; *Zygophyllum berenicense* Schweinf.; *Zygophyllum coccineum* L.; *Zygophyllum coccineum* var. *berenicense* Muschl.; *Zygophyllum desertorum* Forsk.; *Zygophyllum propinquum* Decne.; *Zygophyllum desertorum*; *Zygophyllum coccineum* var. *coccineum* L.

**Common name:** Weißes Jochblatt (DE); White Bean-caper (EN)

**Local Names in the Farasan Islands:** Harm, Rotreyt, and Batbat.

**Geographic distribution range:**

The most widespread *Tetraena* species in Egypt and Saudi Arabia, occurring near saline and sandy habitats [84,85].

**Distribution (countries of occurrence):**

Egypt, Eritrea, Gulf States, Kuwait, Lebanon-Syria, Palestine, Saudi Arabia, Sudan, Yemen [54].

**Biology:**

The species is small shrub, perennial, green. The leaves are fleshy with 2-foliolate cylindrical. The color of the flower is white. The fruit shape is cylindrical [75].

**Flowering and Fruiting period:**

The flowering time starts from October through to November [13].

**Reproduction:**

Seed

**Habitat and Ecology:**

The habitat of this species is coastal and inland on saline sandy soils, sand dunes and plains, and saline depressions [80].

**Population information:**

The populations of *T. coccinea* grow under severe, dry climatic conditions and are stable because they have a good tolerance for these harsh conditions [86].

**Threats:**

Sandy coastal archaeological sites are being lost to coastal developments and damage by vehicle traffic and road works [27]. *Tetraena coccinea* has antimicrobial activity [87], and it is used as a traditional medicine for diabetes, gout, hypertension, and rheumatism [88].

**Criteria applied:**

The regional extent of occurrence (EOO) is 783 km$^2$. The area of occupancy (AOO) is estimated to be 583 km$^2$ (Figure 7). It is vulnerable locally but possible migration in from neighboring countries results in a threat category of Near Threatened. VU is then downlisted to NT because of the presence of populations outside the islands.

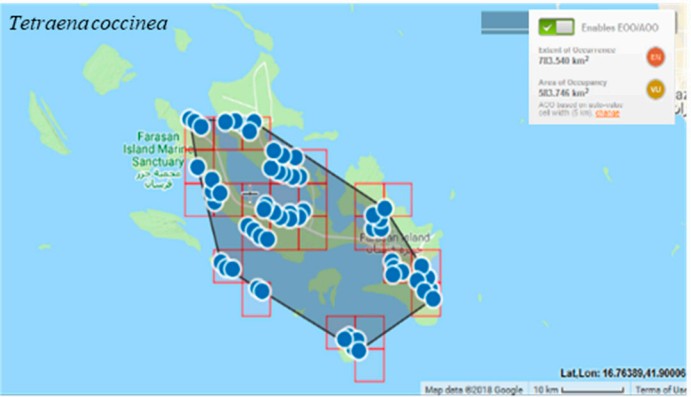
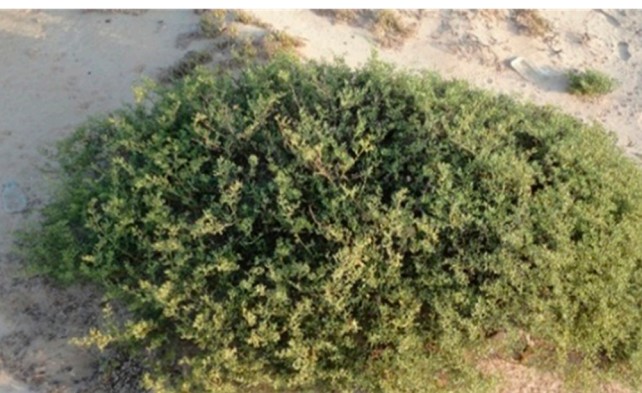

**Figure 7.** Geographical distribution of *Tetraena coccinea* in the Farasan Archipelago (plant photo by R. Al-Qthanin from Faranan Alkabir Island).

B1ab (ii, iii, v) + 2ab (ii, iii, v).
**Previous assessments (and global assessments):**
Has not yet been assessed.
**Conservation actions needed:**
The off-road traffic is the leading contributor to habitat degradation in the main islands, Farasan Alkabir and Sajid, and one of the major threats to species. Management policies need to direct towards controlling off-road vehicle driving, restoring the affected areas, and maintaining ecosystem function in these islands.

*3.6. Tetraena propinqua (Decne.) Ghazanfar and Osborne ssp. migahidii (Hadidi ex Beier and Thulin) Alzahrani*

**Family:** Zygophyllaceae
**Synonyms:** *Zygophyllum migahidii* Hadidi, in Publ. Cairo Univ. Herb. 7 and 8: 328 (1977); *Zygophyllum propinquum* ssp. *migahidii* (Hadidi) Thomas and Chaudhary, Chaudhary in flora of the Kingdom of Saudi Arabia vol. 2: 501 (2001); *Tetraena migahidii* (Hadidi) Beier and Thulin, in Pl. Syst. Evol. 240 (1–4): 36 (2003) [53].
**Common name:** Weißes Jochblatt (DE); White Bean-caper (EN)
**Local Names in the Farasan Islands:** Abu rokaiba (from a label on Rawi and Ilkas 16274), arid, harm (a generic name for several species of *Tetraena*).
**Geographic distribution range:**
The native range is Egypt to Saudi Arabia [54].
**Distribution (countries of occurrence):**
Egypt, Saudi Arabia
**Biology:**
The species is a small shrub, perennial, green. The leaves are fleshy with 2-foliolate with rounded apex. The flowers are white-creamy. The fruit is ovate-oblong to obconical, 5-angled at the upper end [89].
**Flowering and Fruiting period:**
It produces both flowers and fruits, mainly in April–June and Sept–Oct, and occasionally throughout the summer months of July and August [53].
**Reproduction:**
Reproduces by seeds.
**Habitat and Ecology:**
The habitat of this species is coastal and inland on saline sandy soils, sand dunes and plains, and saline depressions.
**Population information:**
The populations of *Tetraena propinqua* ssp. *migahidii* grows under severe dry climatic conditions and it is very limited at Farasan Al-Kabir Island(Figure 8).

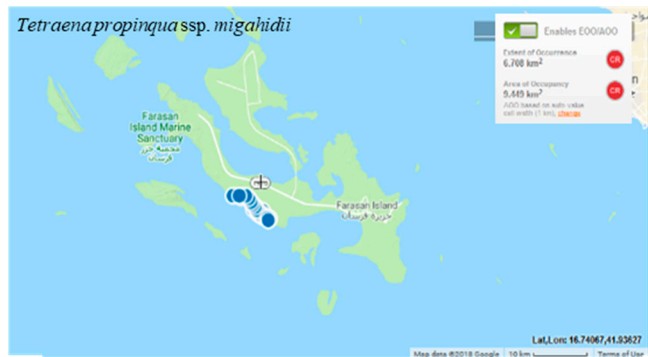
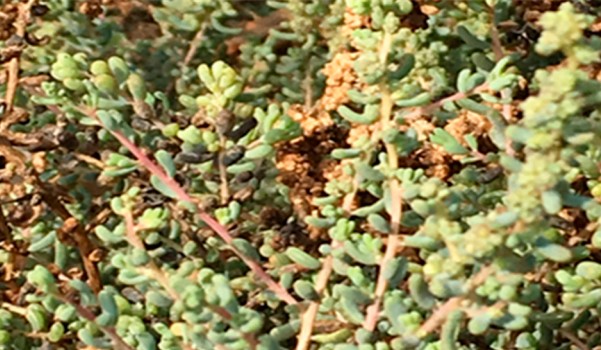

**Figure 8.** Geographical distribution of *Tetraena propinqua* ssp. *migahidii* in the Farasan Archipelago (plant photo by R. Al-Qthanin from Faranan Alkabir Island).

**Threats:**

Many threats exist where *Tetraena propinqua* ssp. *migahidii* is located: continuous land reclamation projects, the construction of roads along the west coast, tourism development [90] and the discarding of fish offal, old nets, and oil drums. Improved access to the seashore through current construction projects could stimulate beach erosion, damage coastal environments, and lead to the further loss of the beaches [69,91], ultimately destroying the habitat of this species.

**Criteria applied:**

The regional extent of occurrence (EOO) and the area of occupancy (AOO) are estimated to be 9 km$^2$. It is critically endangered locally but possible migration in from neighboring countries results in a threat category of endangered. CR is then downlisted to EN because of the presence of populations outside the islands.

B1ab (ii,iii,v) + 2ab(ii,iii,v); C2a(i); D.

**Previous assessments (and global assessments):**

Has not yet been assessed.

**Conservation actions needed:**

The off-road traffic is the leading contributor to habitat degradation in the main islands, Farasan Alkabir and Sajid, and one of the major threats to species. Management policies need to direct towards controlling off-road vehicle driving, restoring the affected areas, and maintaining ecosystem function in these islands.

*3.7. Basilicum polystachyon (L.) Moench*

**Family:** Lamiaceae

**Synonyms:** *Basilicum polystachyon* var. *stereocladum* Briq.; *Lehmannia ocymoidea* Jacq. ex Steud.; *Lumnitzera moschata* (R.Br.) Spreng.; *Lumnitzera polystachyon* (L.) J.Jacq. ex Spreng.; *Moschosma dimidiatum* (Schumach. and Thonn.) Benth.; *Moschosma moschatum* (R.Br.) Druce; *Moschosma polystachyon* (L.) Benth.; *Ocimum dimidiatum* Schumach. and Thonn.; *Ocimum moschatum* Salisb., nom. superfl.; *Ocimum polystachyon* L.; *Ocimum tashiroi* Hayata; *Perxo polystachyon* (L.) Raf.; *Plectranthus micranthus* Spreng.; *Plectranthus moschatus* R.Br.; *Plectranthus parviflorus* R.Br., nom.

**Common name:** Musk basil [92]

**Local Names in the Farasan Islands:** No local name.

**Geographic distribution range:**

Widely distributed in tropical and subtropical climates from tropical Africa to northern Australia [93].

**Distribution (countries of occurrence):**

Angola, Bangladesh, Benin, Bismarck Archipelago, Borneo, Burkina, Burundi, Cambodia, Cameroon, Central African Repu, Chad, China Southeast, Comoros, Congo, Ethiopia, Ghana, Hainan, India, Ivory Coast, Jawa, Kenya, KwaZulu-Natal, Lesser Sunda Is., Madagascar, Malawi, Malaya, Maluku, Mauritius, Mozambique, Myanmar, New Guinea, Niger,

Nigeria, Northern Provinces, Northern Territory, Philippines, Queensland, Rwanda, Saudi Arabia, Solomon Is., Somalia, Sri Lanka, Sudan, Sulawesi, Sumatera, Swaziland, Taiwan, Tanzania, Thailand, Togo, Uganda, Vietnam, Western Australia, Zambia, Zaïre, Zimbabwe [54].

**Biology:**

An annual herb [55].

**Flowering and Fruiting period:**

Flowers and fruits all year round [92].

**Reproduction:**

Reproduces by seeds.

**Habitat and Ecology:**

The *Basilicum polystachyon* thrives primarily in the seasonally dry tropical environment(s) [54]. The plant prefers moist areas like the sides of ditches and streams, hydromorphic habitats, low-lying flooded rice fields, and fallow land [92]. In Farasan Islands, it is growing in the *Vachellia flava* woodlands at 9–12 m.

**Population information:**

The population size of *Basilicum polystachyon* was estimated by Thomas, Al-Farhan, Sivadasan, Samraoui and Bukhari [9] to be between 100–500 individuals. The survey of this species in 2016 and 2017 did not identify any individual, suggesting a possible decline in the population size or probably local extinction.

**Threats:**

The species is under threat from drought, off-road traffic [9], and invasive *Prosopis juliflora*.

**Criteria applied:**

The estimated extent of occurrence (EOO) and the area of occupancy (AOO) are 8 km², which would place the species in the critical endangered CR category according to criterion B. *Basilicum polystachyon* is very rare in the Farasan Islands and known only from the Al Muharraq area in Farasan Alkabir Island (Figure 9) (one location; CR). The decline in the population size is inferred here due to the small population size reported in [9] and the difficulty to find any individual during three visits to the area: April and December 2016, and December 2017. Thus, *Basilicum polystachyon* was initially assessed as CR. However, the probability of the species re-colonizing the islands from outside the region is likely due to the proximity of the Farasan Islands to Africa. Therefore, the preliminary regional category was downlisted to EN.

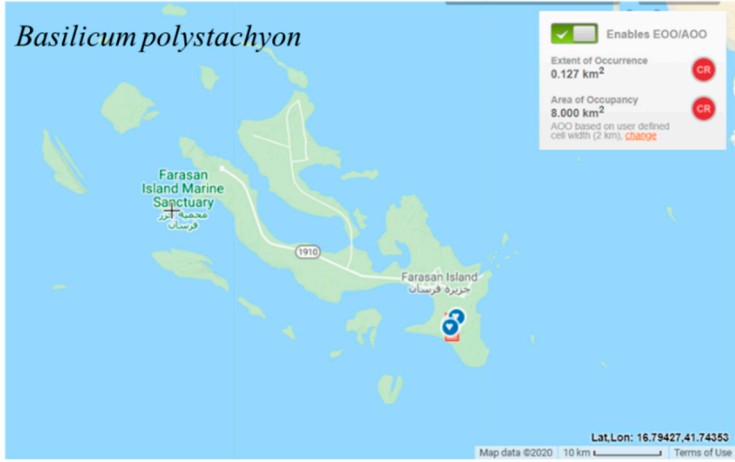
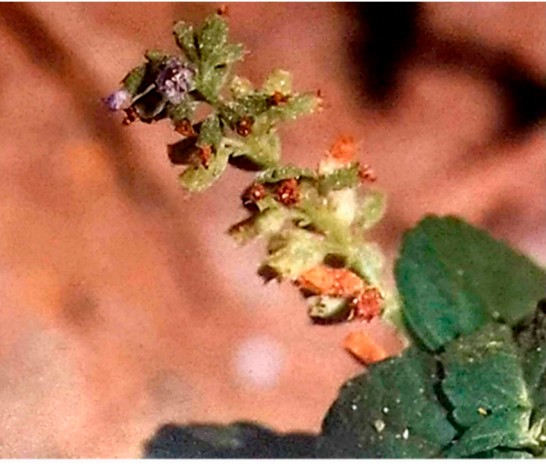

**Figure 9.** Geographical distribution of *Basilicum polystachyon* in the Farasan Archipelago. Plant photo reproduced with permission from Collenette [7].

B1ab (v)+2ab (v)

**Previous assessments (and global assessments):**

Has not yet been assessed.

**Conservation actions needed:**

The Al-Muharraq area should hold special importance in conservation terms, as it is a concentrated pool of the endangered species and occupied by the *Vachellia flava* woodland, the major food of the gazelle [31]. Conservation action in this area must focus on removing the *Prosopis juliflora* which has already invaded the *Vachellia* woodland and negatively affected the native plants in the Farasan Islands [30,94]. *Basilicum polystachyon* and its seeds need to be conserved in the national botanic gardens and seed bank.

*3.8. Dinebra somalensis (Stapf) P.M.Peterson and N.Snow*

**Family:** Poaceae

**Synonyms:** *Drake-brockmania somalensis* Stapf; *Eleusine somalensis* Hack.

**Common name:** No known name.

**Local Names in the Farasan Islands:** No known name.

**Geographic distribution range:**

The species is endemic to the Somalia Masai regional center of endemism [95].

**Distribution (countries of occurrence):**

Tanzania to Northeast Africa (Djibouti, Ethiopia, Somalia, Sudan,) and Saudi Arabia (Farasan Islands) [54,95].

**Biology:**

Mat-forming annual grass spread by stolons [95].

**Flowering and Fruiting period:**

From June to February.

**Reproduction:**

By seed.

**Habitat and Ecology:**

Occupies seasonally flooded locations in silty and saline soils in the seasonally dry tropical biome(s) [54,95]. In the Farasan Islands, growing in a salty clay pan at 5–13 m [7].

**Population information:**

The population of *Dinebra somalensis* has estimated to be between 100–500 individuals [9]. The survey of this species in 2016 and 2017 did not identify any individual, suggesting a possible decline in the population size or probably local extinction.

**Threats:**

It is under numerous threats, especially off-road driving [9], drought, infrastructure development, and urbanization.

**Criteria applied:**

According to criterion B, the estimated area of occupancy (AOO) meets the threshold of the EN category, which was <500 km$^2$ (16.00 km$^2$). The species is rare in Farasan Archipelago and has a restricted geographical distribution to only one island (Farasan Alkabir), the only known Arabian locality (Figure 10). It has been recorded in two locations northwest of the Farasan village, among *Salvadora persica* trees and in abandoned field [7] (one location CR). The population is small, with an estimated 100–500 individual plants [9]. No individuals were observed during field surveys in 2016–2017, indicating a possible decrease in population size due to habitat threats. Thus, *Dinebra somalensis* is assessed as EN; however, the species is likely to re-colonize the Farasan Islands from Africa due to the proximity. Thus, the assessment was downlisted to VU.

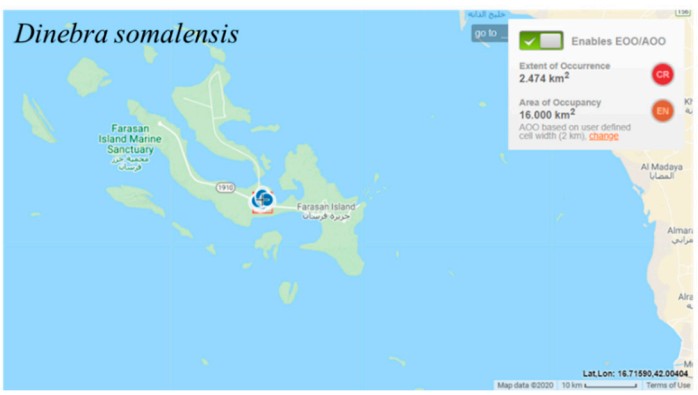
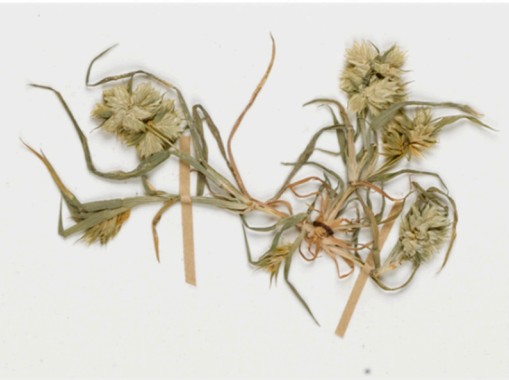

**Figure 10.** Geographical distribution of *Dinebra somalensis* in the Farasan Archipelago. Plant photo reproduced from RBGE herbarium catalogue https://data.rbge.org.uk/search/herbarium/ accessed on 1 June 2019. (*I.S. Collenette-5009* from Farasan Islands).

B1ab (I, ii, iii, v) + 2ab (i, ii, iii, v)
**Previous assessments (and global assessments):**
Has not yet been assessed.
**Conservation actions needed:**
The off-road traffic is the leading contributor to habitat degradation in the main islands, Farasan Alkabir and Sajid, and one of the major threats to *Dinebra somalensis*. Management policies need to be directed towards controlling off-road vehicle driving, restoring the affected areas, and maintaining ecosystem function in these islands.

*3.9. Euphorbia collenetteae D.Al-Zahrani and El-Karemy*

**Family:** Euphorbiaceae
**Common name:** Marar, Saab, Scharath
Local Names in the Farasan Islands: Marar, Saab, Scharath
**Geographic distribution range:**
Endemic to the Red Sea region [57].
**Distribution (countries of occurrence):**
Saudi Arabia (Farasan Islands), Sudan (Port Sudan) and Eritrea (Archico Bay) [57].
**Biology:**
Spiny succulent shrub.
**Flowering and Fruiting period:**
From March to June [57].
**Reproduction:**
By seeds.
**Habitat and Ecology:**
Grows in the desert or dry shrubland biome in the cracks and faults of the fossil coral rocks and on basalt outcrops at 0.5–75 m [54,57].
**Population information:**
*Euphorbia collenetteae* grows mainly in Farasan Alkabir Island, particularly the arid plain of the NW plateau, and on Sajid Island, where it forms scattered clumps. It is very rare in Dumsuk and Dawshak Islands. In 2016 and 2017, the archipelago population was estimated to be around 500 shrubs.
**Threats:**
*Euphorbia collenetteae* is under the threat of habitat loss due to infrastructure development, expansion of cultivation, and urbanization.
**Criteria applied:**
The estimated extent of occurrence (EOO) is 831.329 km$^2$ and the area of occupancy (AOO) is 216.0 km$^2$ (Figure 11), both of which qualify EN according to criterion B. The number of locations is five (NW and SW Farasan Alkabir Island, Sajid Island, Dumsuk

Island, and Dawshak Island). The number of mature individuals was decreasing due to removal of plants for building construction and the establishment of farms on Farasan Alkabir and Sajid Islands. The species is thus initially assessed as EN. The probability of re-colonization on the islands from the small populations in Sudan and Eritrea is very low because *Euphorbia* seeds are mainly ant-dispersed (following ballistic capsule dehiscence). This strongly limits dispersal distances and promotes geographic isolation [96]. Therefore, the preliminary category is left unchanged.

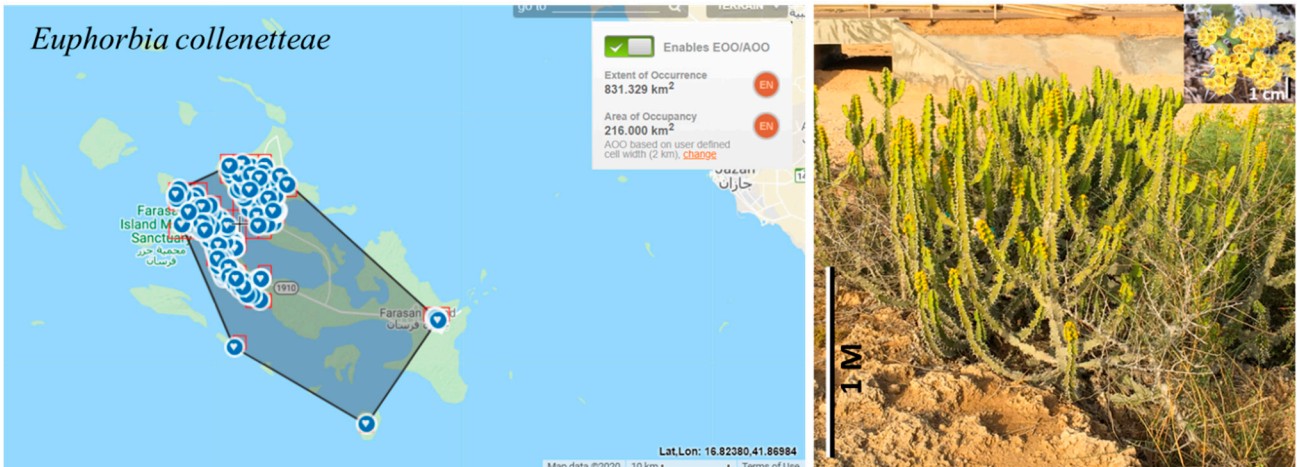

**Figure 11.** Geographical distribution of *Euphorbia colenetteae* in the Farasan Archipelago. Photo by S. Alharbi from Farasan Alkabir Island.

B1ab (i, ii, iii, v) + 2ab (i, ii, iii, v)
**Previous assessments (and global assessments):**
Has not yet been assessed.
**Conservation actions needed:**
The conservation policies in the Farasan Islands should be broadened to include terrestrial plants and land use. *Euphorbia colenetteae* population in Farasan Alkabir and Sajid Islands may have to receive a higher priority in future protection before it is too late.

### 3.10. Micrococca mercurialis (L.) Benth.

**Family:** Euphorbiaceae
**Synonyms:** *Claoxylon mercuriale* (L.) Thwaites; *Mercurialis abyssinica* Hochst. ex Pax and K.Hoffn.; *Mercurialis alternifolia* Lam.; *Microstachys mercurialis* (L.) Dalzell and A.Gibson; *Tragia mercurialis* L.
**Common name:** No known name.
**Local Names in the Farasan Islands:** No known name.
**Geographic distribution range:**
Tropical and S. Africa, Madagascar, Arabian Peninsula, India to Peninsula Malaysia [54].
**Distribution (countries of occurrence):**
Angola, Bangladesh, Benin, Botswana, Burkina, Cameroon, Central African Repu, Chad, Congo, Equatorial Guinea, Ethiopia, Gabon, Ghana, Guinea, Guinea-Bissau, India, Ivory Coast, Kenya, Laccadive Is., Liberia, Madagascar, Malawi, Malaya, Maldives, Mauritania, Mozambique, Myanmar, Nigeria, Saudi Arabia, Senegal, Sierra Leone, Sri Lanka, Sudan, Tanzania, Thailand, Togo, Uganda, West Himalaya, Yemen, Zambia, Zaïre, Zimbabwe [54].
**Biology:**
Annual herb or subshrub.
**Flowering and Fruiting period:**
Rainy season [97].
**Reproduction:**

**Habitat and Ecology:**

Growing in open places in woodlands and bushlands, along rivers and shores, commonly in ruderal habitats, and sometimes as a weed from the sea-level up to 1700 m in altitude [98]. On the Farasan Islands, it grows in damp sand among the palm trees at 9–12 m [7].

**Population information:**

*Micrococca mercurialis* has only been recorded in one location, palm orchard in the Al Muharraq area of Farasan Alkabir Island [7]. No individuals were observed during our field trips to the area. However, Thomas, Al-Farhan, Sivadasan, Samraoui and Bukhari [9] estimated the population to be between 100 and 500 plants.

**Threats:**

*Micrococca mercurialis* has the threat of off-road traffic [9].

**Criteria applied:**

The estimated extent of occurrence (EOO) and the area of occupancy (AOO) are 4 km² and found in one location with a small population (Figure 12); the decline in population was inferred due to the difficulty in observing any individuals in 2016 and 2017. Thus, *Micrococca mercurialis* assessed as CR according to Criterion B. Nevertheless, the probability of species re-colonization from Yemen is likely due to the proximity, and therefore the assessment has been downlisted to EN.

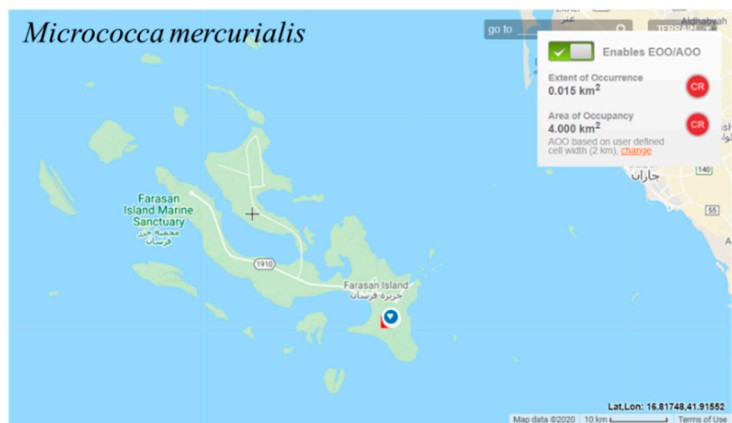 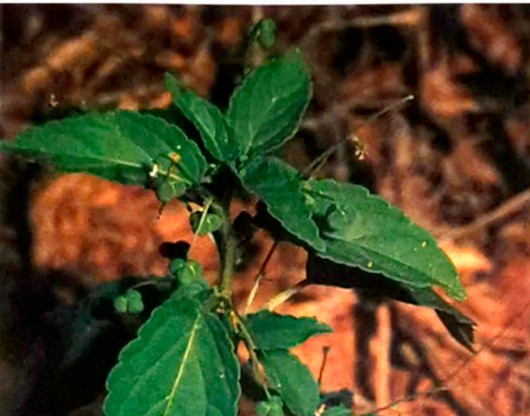

**Figure 12.** Geographical distribution of *Micrococca mercurialis* in the Farasan Archipelago. Plant photo reproduced with permission from Collenette [7].

B1ab (v) + 2ab (v)

**Previous assessments (and global assessments):**

Has not yet been assessed.

**Conservation actions needed:**

Same as those outlined for *Basilicum polystachyon*, as they grow in the same region (the Al Muharraq in Farasan Alkabir Island).

### 3.11. Rorida brachystyla (Deflers ex Franch.) Thulin and Roalson

**Family:** Cleomaceae

**Synonyms:** *Cleome brachystyla* Deflers ex Franch; *C. noeana* Boiss. subsp. *brachystyla* (Deflers ex Franch.) D.F.Chamb. and Lamond; *C. fimbriata* Vicary subsp. *brachystyla* (Deflers ex Franch.) Govaerts.

**Common name:** No known name.

**Local Names in the Farasan Islands:** No known name.

**Geographic distribution range:**

Restricted to the horn of Africa and southern Arabian Peninsula.

**Distribution (countries of occurrence):**

Saudi Arabia (Farasan Islands), Yemen, Djibouti, and Somalia [59].

**Biology:**

Woody, stemmed, bushy, glandular hairy herb [7].

**Flowering and Fruiting period:**
November to March
**Reproduction:**
Reproduces by seeds.
**Habitat and Ecology:**
Growing in deserts or semideserts on gravelly or rocky ground at 10–1000 m [59]. On the Farasan Islands, it can be found growing in cracks on cliffs in deep fossil coral ravines at 9 m in altitude [7].
**Population information:**
The population size estimated to be between 100–500 plants [9]. No individuals were found during the field trips in 2016 and 2017.
**Threats:**
*Rorida brachystyla* is under the threat of habitat loss [9] and drought.
**Criteria applied:**
The species is very rare on the Farasan Islands with a restricted geographical range to one locality (Khallah Bay on Farasan Alkabir Island) (Figure 13). The population size is small, which is estimated to be between 100–500 individuals [9] and likely to decrease due to the threats to the habitat. The estimated extent of occurrence (EOO) and the area of occupancy (AOO) are 12 km². Thus, this species is assessed as EN according to criterion B; however, the likelihood of species immigration from Yemen is probable. Therefore, *Rorida brachystyla* has been downlisted to VU.

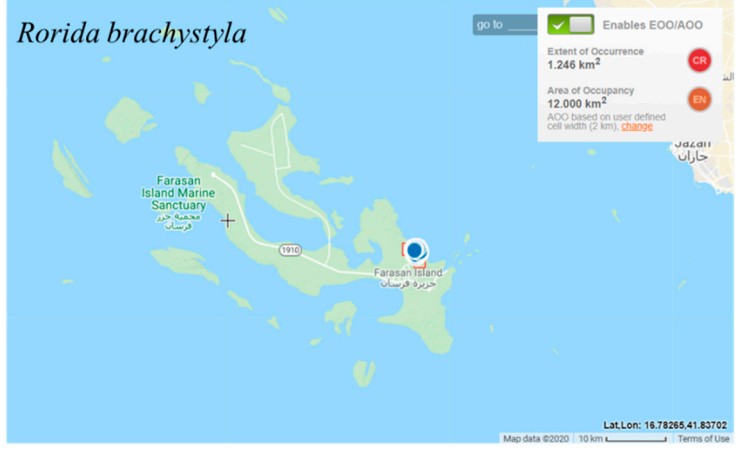 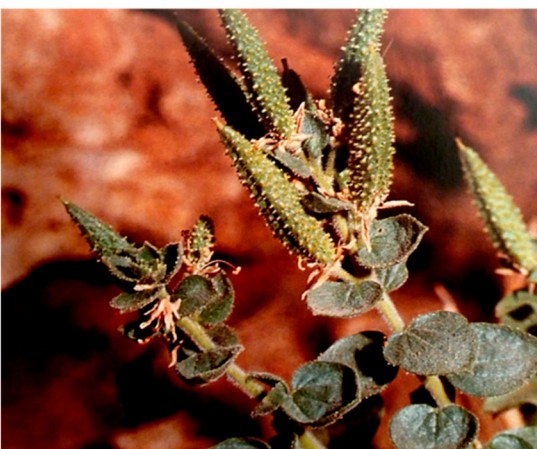

**Figure 13.** Geographical distribution of *Rorida brachystyla* in the Farasan Archipelago. Plant photo reproduced with permission from Collenette [7].

B1ab (i, ii) + 2ab (i, ii)
**Previous assessments (and global assessments):**
Has not yet been assessed.
**Conservation actions needed:**
Protect the habitat and maintain the individuals in the plant nurseries to complement the in situ conservation measures.

### 3.12. Vahlia digyna (Retz.) Kuntze

**Family:** Vahliaceae
**Synonyms:** *Bistella digyna* (Retz.) Bullock; *Haloragis jerosioides* Perrier; *Oldenlandia decumbens* Spreng.; *Oldenlandia digyna* Retz.; *Oldenlandia sessiliflora* Sm.; *Vahlia menyharthii* Schinz; *Vahlia ramosissima* A. DC. ex DC.; *Vahlia sessiliflora* DC.; *Vahlia viscosa* Roxb.
**Common name:** No known name.
**Local Names in the Farasan Islands:** No known name.
**Geographic distribution range:**
Widely distributed from India to tropical Africa [55].

**Distribution (countries of occurrence):**

Botswana, Burkina, Chad, Egypt, Ethiopia, Guinea-Bissau, India, Iraq, Kenya, Madagascar, Mali, Mauritania, Mozambique, Nigeria, Pakistan, Saudi Arabia, Senegal, Sudan, Tanzania, Zambia, Zimbabwe [54].

**Biology:**

Annual herb.

**Flowering and Fruiting period:**

November to July

**Reproduction:**

Reproduces by seeds.

**Habitat and Ecology:**

Growing in the seasonally dry tropical biome [54]. On the Farasan Islands, it can be found in the clay pan in *Vachellia* woodland at 6 m [7].

**Population information:**

The survey during the course of this study (2016 and 2017) did not identify any populations, however, the population size has been estimated in 2010 to be between 100 to 500 individuals [9].

**Threats:**

*Vahlia digyna* is under numerous threats, especially drought, off-road traffic [9], and invasive plants.

**Criteria applied:**

*Vahlia digyna* is a regionally rare species known from only on location in Farasan Islands (the *Vachellia* woodlands in the Al Muharraq area on Farasan Alkabir Island)(Figure 14). The estimated extent of occurrence (EOO) and the area of occupancy (AOO) are 8 km². The *Vachellia* woodland in Al Muharraq has been intensively invaded by the exotic tree *Prosopis juliflora*, which has a negative impact on the native plants in this area [30,94]. The survey conducted by this study did not identify any individuals, suggesting a possible decline in the population size. Therefore, *Vahlia digyna* has been assessed as CR according to criterion B. The probability of re-colonization from outside the region is likely. Thus, the assessment has been downlisted to EN.

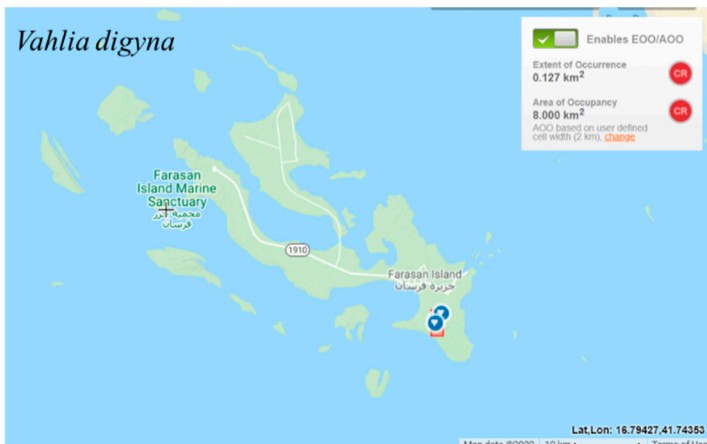
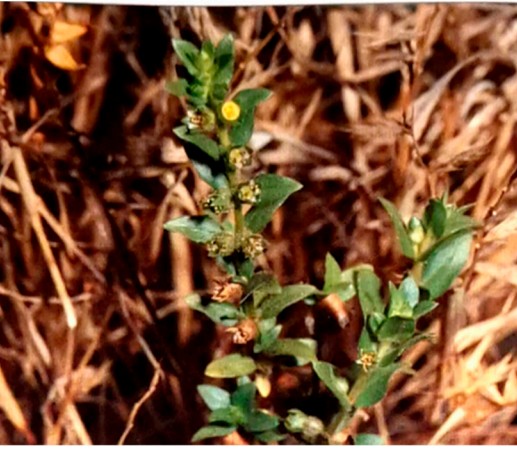

**Figure 14.** Geographical distribution of *Vahlia digyna* in the Farasan Archipelago. Plant photo reproduced with permission from Collenette [7].

B1ab (v)+2ab (v)

**Previous assessments (and global assessments):**

Has not yet been assessed.

**Conservation actions needed:**

Same as those outlined for *Basilicum polystachyon* and *Micrococca mercurialis* as they grow in the same region (the Al Muharraq in Farasan Alkabir Island).

Several limitations and issues may impact the Red List process, such as taxonomic uncertainty, the shortfall in threat knowledge for a given taxon, restricted availability of relevant data, and the lack of regular population trend data. However, in our case, taxonomic dynamism does not present a problem to the red listing process because this study carried out intensive reviews of the Farasan Archipelago flora, including the vulnerable twelve species. In addition, field surveys and collaborating with stakeholders (local people, herbalists, local and visiting foreign botanists) has supported the accurate conservation assessment and IUCN rating.

## 4. Conclusions

These twelve assessments illustrate the extinction risk for nationally rare species and the coastal flora of the Farasan Archipelago. The IUCN list reports that six species have been categorized as endangered, four species as vulnerable, and two species as near threatened at the regional level (Table 2). The categories for the whole species except *E. collenetteae* were upgraded by one level, because these species are not endemic to the islands, which is inconstant with Rodríguez et al. [44] and their modified Regional Red List assessments. The widespread changes in the coastal area, whether intentionally or not, especially of the beach vegetation, have prompted great concern about the conservation of the local biodiversity, which may suffer a decline due to the growth of international and domestic tourism, which is one of the most promising parts of the kingdom's diversification efforts with its Vision 2030 plans [37].

**Table 2.** Red listing status of the twelve taxa including the taxon name and the distribution of species in the Farasan archipelago: Farasan Alkabir (1), Sajid (2), Zifaf (3), Dawshak (4), Qummah (5), and Dumsuk (6). Endemism status in the islands, national IUCN category, criteria applied and global IUCN category was also noted.

| Species | Distribution In the Farasan Islands | Endemism status | National IUCN Category | IUCN Criteria | International IUCN Category |
|---|---|---|---|---|---|
| *Avicennia marina* | 1,2,3 | No | VU° | B1ab(i,ii,iii,v) + 2ab(i,ii,iii,v) | LC |
| *Rhizophora mucronata* | 1,3 | No | VU° | B1ab(i,ii,iii,v) + 2ab(i,ii,iii,v) | LC |
| *Tetraena simplex* | 1,2,3 | No | NT° | B1ab(ii,iii,v) + 2ab(ii,iii,v) | NE |
| *Tetraena alba* var. *alba* | 1,4 | No | EN° | B1ab(ii,iii,v) + 2ab(ii,iii,v); C2a(i);D | NE |
| *Tetraena coccinea* | 1,2,3,4,5 | No | NT° | B1ab(ii,iii,v) + 2ab(ii,iii,v) | NE |
| *Tetraena propinqua* ssp. *migahidii* | 1 | No | EN° | B1ab(ii,iii,v) + 2ab(ii,iii,v); C2a(i); D | NE |
| *Basilicum polystachyon* | 1 | No | EN° | B1ab (v) + 2ab (v) | NE |
| *Dinebra somalensis* | 1 | No | VU° | B1ab (i, ii, iii, v) + 2ab (i, ii, iii, v) | NE |
| *Euphorbia collenetteae* | 1,2,4,6 | No | EN | B1ab (i, ii, iii, v) + 2ab (i, ii, iii, v) | NE |
| *Micrococca mercurialis* | 1 | No | EN° | B1ab (v) + 2ab (v) | NE |
| *Rorida brachystyla* | 1 | No | VU° | B1ab (i, ii) + 2ab (i, ii) | NE |
| *Vahlia digyna* | 1 | No | EN° | B1ab (v) + 2ab (v) | NE |

VU = vulnerable, NT = near threatened, EN = endangered, LC = least concern, NE = has not yet been assessed. Categories marked with (°) are the downlisted ones.

The targeted species are suffering from urbanization, invasive species, overgrazing, increasing infrastructure development, and pollution that are the most threatening factors. Therefore, the conservation of the nationally rare species and the coastal zone in the islands

is required, because they are more vulnerable to natural and unnatural pressure. It is important that conservation policies be broadened to include plants, animals, and land use and to increase awareness, education, and eco-tourism to reduce threats to the Archipelago.

**Author Contributions:** Conceptualization, R.N.A.-Q. and S.A.A.; methodology, R.N.A.-Q. and S.A.A.; software, R.N.A.-Q. and S.A.A.; validation, R.N.A.-Q. and S.A.A.; investigation, R.N.A.-Q. and S.A.A.; resources, R.N.A.-Q. and S.A.A.; data curation, R.N.A.-Q. and S.A.A.; writing—original draft preparation, R.N.A.-Q. and S.A.A.; writing—review and editing, R.N.A.-Q. and S.A.A.; visualization, R.N.A.-Q. and S.A.A.; supervision, R.N.A.-Q. and S.A.A.; project administration, R.N.A.-Q. and S.A.A.; funding acquisition, R.N.A.-Q. and S.A.A. All authors have read and agreed to the published version of the manuscript.

**Funding:** This research was funded by King Khalid University, Umm Al-Qura University and the Saudi Cultural Bureau in London.

**Institutional Review Board Statement:** Not applicable.

**Informed Consent Statement:** Not applicable.

**Data Availability Statement:** Not applicable.

**Conflicts of Interest:** The authors declare no conflict of interest.

## Appendix A

Review of nationally rare species known so far from the Farasan Islands. Grey shaded cells = species found in other localities of Saudi Arabia.

| No. | Species Name | Family | Authors Recorded Species in Another Locality in Saudi Arabia |
|---|---|---|---|
| 1 | *Basilicum polystachyon* (L.) Moench | Lamiaceae | |
| 2 | *Dinebra retroflexa* (Vahl) Panzer | Poaceae | [18] |
| 3 | *Dinebra somalensis* (Stapf) P.M.Peterson and N.Snow (Syn. *Drake-Brockmania somalensis*) | Poaceae | |
| 4 | *Euphorbia collenetteae* D.Al-Zahrani and El-Karemy | Euphorbiaceae | |
| 5 | *Ficus populifolia* Vahl | Moraceae | [17] |
| 6 | *Flueggea leucopyrus* Willd. | Phyllanthaceae | [12] |
| 7 | *Indigofera semitrijuga* Forssk. | Fabaceae | [12] |
| 8 | *Ipomoea hochstetteri* House | Convolvulaceae | [20] |
| 9 | *Limonium cylindrifolium* (Forssk.) Verdc | Plumbaginaceae | [16] |
| 10 | *Micrococca mercurialis* (L.) Benth. | Euphorbiaceae | |
| 11 | *Nothosaerva brachiata* (L.) Wight | Amaranthaceae | [17] |
| 12 | *Rorida brachystyla* (Deflers ex Franch.) Thulin and Roalson (Syn. *Cleome noeana* ssp. *brachystyla*) | Cleomaceae | |
| 13 | *Taverniera cuneifolia* (Roth) Arn. | Fabaceae | [12] |
| 14 | *Vahlia digyna* (Retz.) O. Kuntze | Vahliaceae | |

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
