# Peer review of "Regional Conservation Assessment of the Threatened Species: A Case Study of Twelve Plant Species in the Farasan Archipelago"

_conservation, doi:10.3390/conservation3010011_

Round 1

Reviewer 1 Report

Dear Authors,

Your work on the regionally threatened plant species from the Farasan Archipelago is a crucial piece of research. Your manuscript is very well presented, and I enjoyed reading it. However, I found several major issues in your manuscript.

·         Your field study was almost five years ago (2016/2017). Therefore, your proposed IUCN status does not reflect recent changes in the population trends of those twelve species of plants. Additionally, your published literature data dates back as old as 1971. It is unclear from your manuscript if you used those old records as your georeferencing points. If so, the assumption is those plants are still there, which might not be true.

·         More importantly, the convex polygon of your EOO and AOO assessment of six species (Avicennia marina, Rhizophora mucronate, Tetraena simplex, Tetraenacoccinea, Tetraena alba var. alba, Euphorbia collenetteae) included both the land and waterbody (not only the coastline but also the actual waterbody) regardless the fact that at least four of them are entirely terrestrial species, and none of them are completely aquatic species. Therefore, you are overestimating the EOO and AOO of those species by including waterbodies in your convex polygon. And based on that overestimation, you are downgrading the threatened status which is misleading.

Here, I share some of my thought on how those two issues could be solved:

·         Please collect recent distribution data through field surveys and/or from current publications.

·         Get the convex polygon from the GeoCAT software and mask it in ArcGIS similar (QGIS for example) software. Then extract the land area and exclude the area of the waterbodies to get the AOO.

Author Response

Hello Dear, 

We really appreciate for reviewing the paper and we work through that comments as we can 

many thanks 

Reviewer 2 Report

The paper entitled: "Regional conservation assessment of the threatened species: a case study of twelve plant species in the Farasan Archipelago", by Al-Qthanin and Alharbi, submitted to Conservation, falls within the scope of the journal, and treat the assessment of the threatened species in Farassan Archipelago. the findings of the study are of a great importance as they highlight some species assessed for the first time. Also, the paper gives a new classification of two species Avicennia marina and Rhizophora mucronata, as highly threatened, according to the IUCN assessment guidelines. 

The paper is well-written and well-structured, some minor revisions are needed before considering the paper for publication.

* Giving the morphological characteristics of the listed plants would be very helpful in term of conservation.

* In Table 1, please note that the scientific name of the specie Avicennia marina (Forssk.) Vierh. remained the same (WFO (2022): Avicennia marina (Forssk.) Vierh. Published on the Internet; http://www.worldfloraonline.org/taxon/wfo-0000303022. Accessed on: 19 Oct 2022); same for Rhizophora mucronata Poir. (WFO (2022): Rhizophora mucronata Poir. Published on the Internet;http://www.worldfloraonline.org/taxon/wfo-0001131556. Accessed on: 19 Oct 2022)

** I invite the authors to check the references section, and change the referencing style (it should respect Conservation guidelines).

Author Response

we really appreciate for reviewing paper and we working through comments as we can 

many thanks 

Reviewer 3 Report

In the manuscript “Regional Conservation Assessment of the Threatened Species: A Case Study of Twelve Plant Species in the Farasan Archipelago” the regional IUCN categories and criteria have been used to assess the conservation status of twelve targeted taxa of the Farasan Archipelago based on the data collected during field surveys and a literature review.

Although 1) there is lack of long-term monitoring of the targeted species, 2) the assessment was carried out in 2017, i.e. 5 years ago and 3) the methodology is not adequately explained, regional assessments are important because they make it easier to implement conservation actions in particular areas, which may not have been apparent at the global scale.

Key points that need improvement:

It is very important to enrich your assessment with population data from field surveys, to record and classify the direct threats according to IUCN and to use the latest version of the Guidelines for Using the IUCN Red List Categories and Criteria.

In more detail:

Introduction

In my opinion, more information should be provided about the Farasan archipelago and perhaps the paleogeography of the area. The reports on the pressures/threats are very extensive and I believe they can be reduced. Additionally, information about the plant species under study should be provided in this section and not in the results.

Line 33: Please replace km2 with km2

In Line 34, replace Frasan to Farasan

Line 35: Please mention these species

Line 43:  You mention that “The significance of the Farasan Islands flora in Saudi Arabia, thus, is not in terms of its endemism, which is low, but in the presence of those ten species rare elsewhere in the Arabian Peninsula”. It is not clear to me whether you are going to study 12 species (as mentioned in the title of the manuscript) or 10.

Lines 53-56 could be deleted

In Line 77 replace “face” with a more appropriate word.

Lines 100-105: more recent literature is available

Material and methods

In line 126 you could mention the species under study.

Lines 127-128 are confusing

In Line 150 I suppose you mean “IUCN criteria and categories version 3.1” and not version 13

Lines 166-167: The Guidelines for Using the IUCN Red List Categories and Criteria is THE most important document and you should use the latest version available at:  https://www.iucnredlist.org/resources/redlistguidelines

The section of Materials and Methods is not well written. It is not clear what did you did in your field study and how did you use the data from herbarium records?

Results and discussion

Lines 338-339: Could you please explain how the AOO was calculated at 7 km2, while for its estimation a 2 × 2 km2 grid was used

For regional assessment:  In my opinion it is more appropriate to follow a different structure (with subsections) in your assessments. For example:

- Taxonomy and nomenclature

- Common name

- Local Names in the Farasan Islands

- Geographic distribution range

- Distribution: Countries of occurrence

-Biology

-Flowering and Fruiting period

- Reproduction

- Habitat and Ecology

- Population information: this is a very important part in the assessment and you should provide all the information about the population size (census data or estimation of the population size during the field survey). Indicate clearly the duration of the monitoring period.

- Threats  

- Criteria applied (in detail)

- Previous assessments (and global assessments)

- Conservation actions needed

- Figures (figure of the species and its distribution range)

It is very important to record and classify the direct threats that have impacted, are impacting or may impact the status of the species under study according to IUCN (Threats classification Scheme)

Moreover, Population data from field surveys are not presented. Considering the importance of population size estimates for predicting long-term population survival, it is important to provide this information

Figures

In Figure 1, please replace “most pollution area” with “most polluted area”

In Appendix, Figures 2, 3 and 4 must be removed

Please check for grammatical and typographical errors throughout the manuscript

Author Response

(The authors gave the same response as above.)

Round 2

Reviewer 1 Report

Dear authors,

Your revised manuscript is very difficult to follow. I am not sure if you uploaded your final version. It shows all the editing you have done using track change, and the format is very unusual. It looks like a draft of a manuscript with an enabled track change option. Please double-check.

Author Response

many thanks 

Reviewer 3 Report

Dear Authors,

Thank you very much for your reply.

The manuscript has highly improved, although I think that you could reconsider to list the threats according to the IUCN (I mean to provide the IUCN threat code)

Author Response

Many thanks for your comments

I do not exactly understand what you mean to the list, but we tried to follow the IUCN categories as we can with our data.